# RECURSIVE REASONING WITH NEURAL NETWORKS

**Jonas Jürß**[*] **& Dulhan Jayalath**[*]
Department of Computer Science and Technology
University of Cambridge
Cambridge, CB3 0FD, UK
`{jj570,dhj26}@cl.cam.ac.uk`

## ABSTRACT

Many problems can naturally be thought about recursively. However, neural networks fundamentally cannot reason this way on arbitrarily large problems. This is because they do not have the memory to maintain state for the maximum recursion depth required. Solving this issue would enable neural networks to reason like a wide range of classical recursive algorithms (e.g., tree search in model-based RL). To address this, we propose a neural architecture augmented with a stack that learns to save and recall state as needed. We empirically demonstrate the utility of this method on a recursive neural algorithmic reasoning task (learning depth-first search) and show that our architecture leads to improved generalization.

## 1 INTRODUCTION & BACKGROUND

Reasoning recursively requires memory at least large enough to store as many states as the maximum recursion depth of the problem. Since neural networks have a fixed size, they cannot reason this way on arbitrarily large problems. This is problematic in Neural Algorithmic Reasoning (NAR), where networks learn to execute algorithms by mimicking steps of algorithmic computation (Veličković & Blundell, 2021; Xhonneux et al., 2021; Ibarz et al., 2022), because to execute a recursive algorithm, the network must be able to reason correctly on arbitrarily large problem instances.

To address this fundamental issue, we introduce a neural architecture that incorporates a stack. Inspired by call stacks in computer programs, our stack is used by the network to save or recall state. This enables the network to execute algorithms with arbitrary recursion depth. We demonstrate empirically that our approach is beneficial for neural algorithmic reasoning on depth-first search (DFS)—a classic example of a recursive problem. Our architecture permits neural networks to reason like a recursive algorithm, something that was previously not possible. By doing so, our work is a step towards the goal of building generalist algorithmic learners that can compose basic algorithms into pragmatic solutions (Xhonneux et al., 2021; Ibarz et al., 2022).

## 2 METHOD & EXPERIMENTS

We implement a stack to store and recall state from a neural network as described in Figure 1. The *hints*[1] describe the algorithmic computation state. In DFS, hints include a pointer to the current node being explored, its predecessor, and whether each node has been visited. In theory, predicting hints aligns the network with the reasoning process of the algorithm.

As the prototypical recursive algorithm, we choose depth-first search to demonstrate our method. We modify the CLRS algorithmic reasoning benchmark (Veličković et al., 2022) to implement DFS with hints based exactly on the variables of the algorithm defined in Cormen et al. (2009). We append the stack operation to these hints. Our train, validation, and test sets consist of a mix of Erdős–Rényi (E-R) graphs and random binary trees generated with particularly long chains to evaluate high recursion depth. We use a Graph Neural Network (GNN) as the network component of our method. For complete experimental details, see Appendix A and our implementation.[2]

---

[*]These authors contributed equally.
[1]See Veličković et al. (2022) for a complete description.
[2]https://github.com/DJayalath/gnn-call-stack

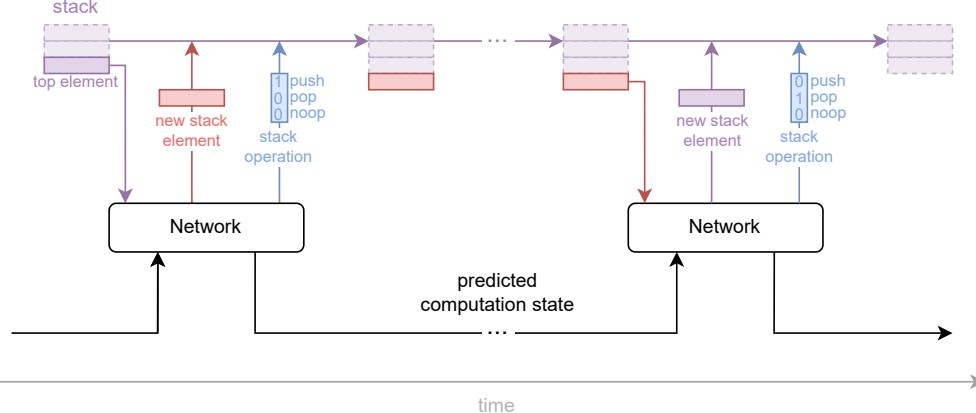

Figure 1: **Augmenting a neural network with a stack.** The input to the network is the state at the top of the stack and the algorithm's computation state (a set of hints). **Inference.** The network predicts the next stack element, stack operation, and next algorithm computation state. If the operation is a push, the predicted next stack element is placed on the stack; if it is a pop, the current top stack element is discarded and so is the next stack element prediction. **Training.** We supervise the stack operation selection (and the rest of the algorithmic computation state) by sampling at steps in the computation of the recursive algorithm. This forms the algorithmic computation state (hints) that are predicted by the network and input to the next step.

**Results.** When we augmented the approach by Veličković et al. (2022) with a stack controlled by a stack hint prediction, we achieved a mean accuracy of $\mathbf{89.1\%} \pm \mathbf{3.0}$. Without the stack (and only the stack hint prediction), the method achieved an accuracy of $60.7\% \pm 4.0$. With neither augmentation, the method by Veličković et al. (2022) achieved an accuracy of $64.0\% \pm 1.2$ using our experimental setup.

## 3    DISCUSSION

We hypothesize that our network is learning to use the stack in the same way as a call stack, saving and restoring state across time. Hence, our method enables reasoning recursively. The improvement in accuracy is then a consequence of the network's ability to reason like the DFS algorithm. Moreover, we avoid the well-known memory bottleneck seen in Recurrent Neural Networks (RNNs) caused by the fixed size of the hidden state in the network (Jurafsky & Martin, 2022, Chapter 9) as we can store states and recall them without loss of information.

Our method resembles that of an RNN or Long Short-Term Memory (LSTM). However, it is important to note the difference. We are able to store the exact past hidden state for an arbitrarily long time and learn to recall it when required. To do so, we must supervise the network's prediction of stack operations. This is trivial for recursive algorithms since call stack usage is well-defined.

## 4    CONCLUSION & FUTURE WORK

We proposed augmenting a neural network with a stack based on the hypothesis that it allows the network to generalize better than previous work on recursive reasoning tasks. We showed that this is indeed the case for neural algorithmic reasoning with depth-first search. This method could be further generalized beyond recursive reasoning to general recursive problems by learning to use a stack without intermediate supervision (e.g., using reinforcement learning). Nevertheless, since the call graph of a recursive algorithm is precisely a depth-first search, we believe that our solution can be applied to reason like any recursive algorithm where this execution path can be known in advance. Our work has enlarged the class of algorithms that we can precisely reason about with neural networks.

ACKNOWLEDGEMENTS

We thank Dobrik Georgiev for advice and input on the CLRS benchmark, Petar Veličković for several insightful discussions during this project, and Yonatan Gideoni for kindly reviewing our draft of this paper.

URM STATEMENT

The authors acknowledge that at least one key author of this work meets the URM criteria of the ICLR 2023 Tiny Papers Track.

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

## A  EXPERIMENT SETUP

We follow the general training setup employed by Ibarz et al. (2022) in their single task experiments. Like Veličković et al. (2022), we apply *teacher forcing* by replacing the predicted computation state of the last time step with the ground truth with 50% probability during training. Crucially, in contrast to the mentioned previous work on NAR, we do not pass a regular hidden state. As compute limits us to relatively small training graphs, it is otherwise easy to overfit to these graphs using this hidden state instead of the stack. Whereas the hidden state would not be able to capture sufficient information for larger graphs, it is easier to learn its usage at the start of training when the stack operations have not yet been learned and we cannot rely on the correct stack element to be present at each time step. Commands to reproduce our results are given in the accompanying repository.

### A.1 DFS IMPLEMENTATION

Note that the original specification of DFS for NAR by Veličković et al. (2022) requires prediction of all node-level features (e.g., the parents of all nodes) at once. This inevitably leads to the memory bottleneck issue described in Section 3. Therefore, we collect the output over different time steps, similar to what an RNN would do. We collect the mapping from all nodes to their parents by taking the predicted current node at each time step and overwriting its parent with the predicted parent. Previous work did not exhibit this memory issue as they passed node-level computation states between the time steps. The reason this works in this particular case is that the specific representation of the hints (in terms of nodes) bounds the depth of the call stack in the case of DFS. However, this does not allow learning recursive reasoning. In order to avoid this issue, we modified most of these computation states to be graph-level so that we did not have global information that avoids the need for a call stack.

### A.2 GRAPH GENERATION

Following Ibarz et al. (2022), we generate graphs on the fly. However, we observe that their random E-R graphs often only require a low recursion depth. Therefore, we substitute 15% of the generated graphs with random binary trees. We use graphs of size 4, 8, 16, 24, and 32 for training whereas only graphs of size 32 are used for validation (with early stopping) and testing.

### A.3 GRAPH-LEVEL CALLSTACK

In our concrete implementation of the general method in Section 2, we employ a Message-Passing Neural Network (MPNN). To utilize a graph-level call stack, we construct a stack element by max-pooling node embeddings.

### A.4 HYPERPARAMETERS

Table 1: **Parameter values in our network configurations.**

| Parameter | Value |
|---|---|
| GNN Architecture | MPNN (Gilmer et al., 2017) |
| Epochs | 15000 |
| Node pooling operation | max |
| Hidden state dimension | 128 |
| Stack embedding dimension | 64 |
| Activation | ReLU |

## B ABLATION

In Figure 2, we perform an ablation where we compare the training curves of three configurations. Adding a hint for the stack operation and allowing the network to learn a stack as described in Section 2 leads to a significant improvement in test accuracy. Only learning the additional stack operation hint but not using a stack does not yield a similar improvement. This indicates that the crucial element is the stack memory itself rather than an inductive bias induced by the new hint.

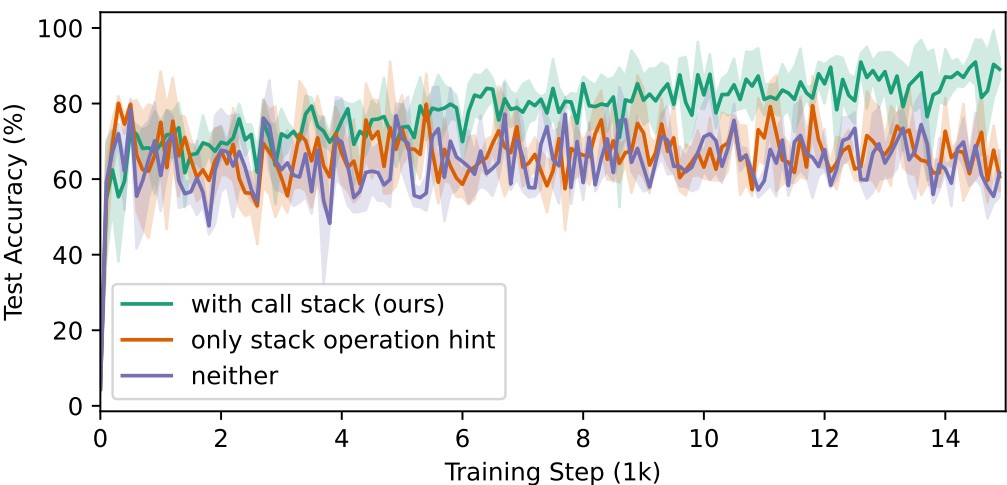

Figure 2: **A call stack is advantageous for learning DFS.** We show the means of three runs with different random seeds. The shaded area shows the standard deviation. "with call stack" describes our contribution which is the incorporation of a call stack with a stack operation hint as described in Figure 1. "only stack operation hint" is the same but without inputting the stack elements to the network (i.e., we still predict the stack operations, but do not make use of the stack itself). "neither" removes both the stack and the stack operation hint, leaving only the hints based on the variables of the algorithm.

