# OpenReview forum: "Recursive Reasoning with Neural Networks"
_ICLR.cc/2023/TinyPapers — Submitted to Tiny Papers @ ICLR 2023_

### Official Review · Reviewer_tdYr · 2023-04-01

**Confidence:** 4

**Summary Of Contributions:**

In this submission, the authors consider the problem of applying neural networks (a GNN in this case) to recursive problems. They introduce a neural architecture that incorporates a stack. They provide a prototypical experiment on learning to depth-first-search.

**Rating:**

Clear, Correct, and Reproducible (CCR): a submission which meets the reviewing criteria

**Strengths And Weaknesses:**

Strengths:
- Considering the training of neural networks on recursive or inductive problems is an important topic
- In their prototype exerpiment, a neural network with a stack improves the mean accuracy from 60.7% to 89.1%

Weaknesses:
- The authors consider one architecture (GNNs) and one recursive algorithm (DFS), but the title and introduction are way to broad.
- The stack operation selection is supervised

**Suggested Changes:**

The title and introduction are way too broad. I would recommend the authors specify this.

---

### Official Review · Reviewer_v2uq · 2023-04-01

**Confidence:** 3

**Summary Of Contributions:**

This work introduces a simple strategy to algorithmic reasoning by incorporating a stack to exactly save and recall the network's states.

**Rating:**

Great Start (GS): a submission which meets some of the reviewing criteria but has room for improvement

**Strengths And Weaknesses:**

Strengths
- This paper studies an important research question of algorithmic reasoning
- The proposed method is simple and yet effective

Weaknesses
- The writing could be improved, while the high level idea of the proposed method is clear, many of its implementation details are missing. Particularly, how did the authors modify the network to interact with the stack, i.e. as the stack size is always changing, what is the last layer of the network?
- Is it fair to compared to the non-stack variant as this method requires call stack hints? What is the complexity overhead when introducing the stack?
- In Appendix A, "replacing the predicted computation state of the last time step with the ground truth with 50% probability during training" is commonly referred to as "scheduled sampling" [A] rather than teacher forcing.

[A] Bengio, Samy, et al. "Scheduled sampling for sequence prediction with recurrent neural networks." Advances in neural information processing systems 28 (2015).

**Suggested Changes:**

See Weaknesses

---

### Official Review · Reviewer_aLDn · 2023-04-01

**Confidence:** 3

**Summary Of Contributions:**

The paper proposes a new neural network architecture which is augmented with a stack to save and restore states across time. This work makes reasoning recursively with neural networks possible and leads to better constructing generalist neural algorithmic learners.

**Rating:**

High Potential (HP): a submission which meets the reviewing criteria and has potential to make an impact on the field

**Strengths And Weaknesses:**

Thank you for submitting this work to Tiny Papers @ ICLR 2023. The paper targets an important aspect of neural networks and presents an interesting solution to incorporate stacks with the architecture.

Strength
- The clarity is very-well satisfied. The problem that neural networks cannot reason recursively is presented clearly in the paper. The proposed architecture, training and testing pipeline, and the results are described explicitly as well. The idea of incorporating stacks with neural networks is very interesting and has certain degree of novelty.
- The results (i.e., nearly 30% increase in accuracy) justifies their hypothesis.
- The authors are aware of the importance of the reproducibility of the experiments and mention that their implementation will be provided in the camera-ready version.
- This paper successfully follows the basic formatting requirements, page limit, and the ICLR code of conduct.

Weakness
- There are several relevant literature and some components of their experiments (e.g., hints and  the CLRS algorithmic reasoning benchmark) mentioned in the paper. However, it would be better to describe the experiment setting (i.e., the hyperparameters, datasets, neural network backbone, etc.) in more detail.
- For the accuracy reported in the result section, I would suggest to state more about what the accuracy means and whether it is possible to compare the experimental results with some published baseline results to make the stated improvement more convincing.
- The paper does claim the goal and importance of this new architecture, but it would be better to describe some concrete applications of it. For example, some specific tasks that current architectures could not/struggle to solve but can be easily addressed by this proposed architecture.

**Suggested Changes:**

The proposed idea on augmenting neural network with stack is innovative and promising to make some impact in the field. Here are some suggestions to further improve the work.
- Explicitly describe the experimental setting (i.e., hyperparameters, neural network backbone, and datasets) and the result accuracy. Adding some reports on the performance (i.e., time and space complexity) would be beneficial.
- Compare the results of the proposed architecture with SOTA architectures.
- For future directions, apply this proposed architecture to solve some tasks that are otherwise difficult for other architectures.

---

> ### Author Response · Authors · 2023-04-15
> **Clarification on time and space complexity**
>
> Thanks for your comments and suggestions on our paper. We have included a general reply in [this response](https://openreview.net/forum?id=TS8l4VS7_BK&noteId=kq0r3BpHyXh). We particularly appreciate your point about referring to time and space complexity and we are keen to address this. Please could you provide some more details about your suggestion. Would you propose adding concrete numbers from our experiments or a more theoretical analysis of the complexity of our training/inference method? In either case, please could you elaborate.

---

### Official Review · Reviewer_5JMp · 2023-04-03

**Confidence:** 2

**Summary Of Contributions:**

The authors propose a novel approach to recursive reasoning with Neural Networks, wherein they augment a DNN with a stack. They demonstrate the performance of their approach on the CLRS algorithmic reasoning benchmark.

**Rating:**

Great Start (GS): a submission which meets some of the reviewing criteria but has room for improvement

**Strengths And Weaknesses:**

Strengths:
- The paper proposes an interesting idea to tackle the problem of algorithmic reasoning
- The method has been described intuitively and clearly
- Ablation has been done to explicate the impact of adding the stack to the training

Weaknesses:
- The writing can be made more self-contained. Currently, there are a lot of references to prior work which makes it a bit harder to understand the impact of the obtained results. For example,
- HPs have not been explicitly reported. I am not sure how reproducible the results are.
- I am missing baselines in this experiment. While it is clear that the performance is being evaluated on a benchmark for recursive reasoning, what is not evident to me is the notion of a good performance for this approach. Further details need to be provided explaining what the results mean quantitatively i.e. what is an expected test accuracy, what has been obtained and how do we explain the difference etc.

**Suggested Changes:**

- I would suggest explaining the connection between expected performances in the CLRS benchmark and how indicative is that performance to the original problem of recursive reasoning. While this has been discussed in the cited paper, I think it makes sense to add a small explanation to make the writing more self-contained
- It has been mentioned at multiple points that the method differs from  RNNs. Can RNN baselines be formulated for this task? If so, I would recommend adding that to the plot. If not, I would recommend indicating the performance of previous approaches on the benchmark to give the reader a notion of what the obtained accuracy means.

---

### Author Response · Authors · 2023-04-15
**Response to the reviewers**

We would like to thank the reviewers for their comments and suggestions. Especially as this is our first time submitting a paper, your comments were particularly helpful to highlight points that we previously missed.

We have updated the submission with changes that address the following points:

Expected accuracy and what explains the difference in accuracy.

- We rewrote the results paragraph to state which work we compare to [A] and provided an explanation in the discussion section of why our method achieves greater accuracy (stronger alignment with the algorithm and no memory bottleneck when recalling state)

Clarity on setting, hyperparameters, and datasets.

- We provide an additional section in Appendix A with hyperparameters. We have additional details on the setting and datasets as well in Appendix A.
- We have also included the link to our code which contains commands to reproduce our results

Teacher forcing with 50% probability during training is commonly referred to as "scheduled sampling” [B].

- We use our terminology for two reasons. Firstly, since we keep the teacher forcing probability constant, this method is not scheduled. Aside from this, we note that we are consistent with the terminology in [A] to avoid confusion.

[A] Veličković, Petar, et al. "The CLRS Algorithmic Reasoning Benchmark." Proceedings of the 39th International Conference on Machine Learning, PMLR 162:22084-22102, 2022.

[B] Bengio, Samy, et al. "Scheduled sampling for sequence prediction with recurrent neural networks." Advances in neural information processing systems 28, 2015.

---

### Meta-Review · Area_Chair_mDmh · 2023-04-04

**Recommendation:** Invite to present
**Confidence:** 3

**Metareview:**

The paper proposes a novel approach to recursive reasoning with Neural Networks, which is an important aspect of neural networks. The method is clearly presented and the results demonstrate its effectiveness. The submission can be improved by providing more details about the experimental setting, hyperparameters, and comparing the results with other state-of-the-art architectures.

**Summary:**

The paper proposes a novel approach to recursive reasoning with Neural Networks by incorporating a stack, and the results demonstrate its effectiveness

**Comments And Feedback To The Authors:**

- The paper could benefit from providing a more detailed explanation of the experimental setting, hyperparameters, and datasets.
- It would be beneficial to compare the performance of the proposed architecture with SOTA architectures to give the reader a notion of what the obtained accuracy means.
- Applying the proposed architecture to solve some tasks that are otherwise difficult for other architectures could be an interesting direction for future work.


**Reason For Not Giving A Higher Recommendation:**

The paper does not compare the performance of the proposed architecture with SOTA architectures, and applies it to solve some tasks that are otherwise difficult for other architectures.

**Reason For Not Giving A Lower Recommendation:**

The method is effective and is clearly presented.

---

### Decision · Program_Chairs · 2023-04-07

Invite to present